# Objective Assessments of Smoking and Drinking Outperform Clinical Phenotypes in Predicting Variance in Epigenetic Aging

**DOI:** 10.3390/genes15070869

**Published:** 2024-07-02

**Authors:** Robert Philibert, Man-Kit Lei, Mei Ling Ong, Steven R. H. Beach

**Affiliations:** 1Department of Psychiatry, University of Iowa, Iowa City, IA 52242, USA; 2Behavioral Diagnostics LLC, Coralville, IA 52241, USA; 3Department of Sociology, University of Georgia, Athens, GA 30602, USA; karlo@uga.edu; 4Center for Family Research, University of Georgia, Athens, GA 30602, USA; tmlong@uga.edu (M.L.O.); srhbeach@uga.edu (S.R.H.B.); 5Department of Psychology, University of Georgia, Athens, GA 30602, USA

**Keywords:** smoking, alcohol, epigenetic aging, DNA methylation, cg05575921

## Abstract

The reliability of the associations of the acceleration of epigenetic aging (EA) indices with clinical phenotypes other than for smoking and drinking is poorly understood. Furthermore, the majority of clinical phenotyping studies have been conducted using data from subjects of European ancestry. In order to address these limitations, we conducted clinical, physiologic, and epigenetic assessments of a cohort of 278 middle-aged African American adults and analyzed the associations with the recently described principal-components-trained version of GrimAge (i.e., PC-GrimAge) and with the DunedinPACE (PACE) index using regression analyses. We found that 74% of PC-GrimAge accelerated aging could be predicted by a simple baseline model consisting of age, sex, and methylation-sensitive digital PCR (MSdPCR) assessments of smoking and drinking. The addition of other serological, demographic, and medical history variables or PACE values did not meaningfully improve the prediction, although some variables did significantly improve the model fit. In contrast, clinical variables mapping to cardiometabolic syndrome did independently contribute to the prediction of PACE values beyond the baseline model. The PACE values were poorly correlated with the GrimAge values (r = 0.2), with little overlap in variance explained other than that conveyed by smoking and drinking. The results suggest that EA indices may differ in the clinical information that they provide and may have significant limitations as screening tools to guide patient care.

## 1. Introduction

Over the past several years, a vast number of studies have shown that numerous cancers and diseases of aging, such as coronary heart disease, diabetes, and Alzheimer’s disease, are associated with the acceleration of epigenetic aging (EA) [1,2,3]. In the United States, this research has led some to begin using EA assessments occasionally in clinical settings. Most frequently, this use occurs in settings such as Concierge Medicine clinics, which are less impacted by managed care. In the managed care settings that dominate American healthcare, the low sensitivity and specificity of EA indices for specific disorders and the lack of third-party reimbursement are prohibitive barriers to the use of these regression-based tools. Nonetheless, to the extent that EA assessments can be shown to indicate risks for specific clinical phenotypes or increase the enthusiasm for screening, they could become useful adjunctive assessments. However, currently, the amount of actionable clinical information that EA assessments convey is unclear. In particular, despite the profligacy of associations with clinical conditions, there is limited consensus as to the identity of the factors that are driving these associations. Further, there are many indices of EA and they may provide different types of actionable information. Finally, as for many other areas of medicine, there are insufficient samples of Black Americans and other minority groups to directly examine whether the patterns of association are similar for them and so whether EA indices can be safely used in similar ways for all patients.

Previous studies have shown that two of the largest and most targetable clinical drivers of EA may be smoking and drinking. For example, a number of groups have shown that the self-reporting of alcohol consumption is associated with accelerated EA [4,5,6,7]. Similarly, self-reported smoking has long been known to be a driver of EA [8,9]. At the same time, in comparison to generally accepted biomarkers such as cotinine, carbohydrate-deficient transferrin, and blood alcohol concentrations, the self-reporting of both smoking and drinking in both general and clinical research settings substantially understates the actual use [10,11,12,13,14,15]. If so, studies that use self-reports of cigarette or alcohol consumption may markedly underestimate the impact of these habits on EA. 

Fortunately, there are alternatives to self-reports that have been provided by the DNA-methylation-based (i.e., epigenetic) assessment of substance use. Specifically, over the past several years, we have shown that the methylation-sensitive digital PCR (MSdPCR) assessment of cg05575921, a CpG motif found in the aryl hydrocarbon receptor repressor (AHRR) gene that is a generally accepted biomarker of smoking intensity, robustly detects daily smoking with an AUC > 0.98 and can be used to assess the effectiveness of smoking cessation therapy [16,17,18,19]. Similarly, the Alcohol T-Score (ATS), a metric that uses input from four separate MSdPCR assays, is a sensitive and specific predictor of heavy alcohol consumption (HAC) [20,21]. In a direct comparison, it outperformed carbohydrate-deficient transferrin (CDT) assessment in predicting HAC, with an overall area under the curve of 0.96 [20]. In prior work, we showed that these two assessments of substance consumption accounted for 57% of the GrimAge acceleration and 95% of the shared variance of accelerated aging between various popular measures of EA [14,22]. 

However, to date, the ability of the ATS and the MSdPCR assessments of cg05575921 (Dcg05575921) to predict accelerated aging in older populations is not well understood. Determining this capacity is important for both clinical and research reasons. From the clinical standpoint, the majority of patients seeking assessments of aging tend to be significantly older, while the rates of diabetes and other diseases of aging are higher than in the 28-year-old subjects from our previous study [14]. Therefore, examining the magnitude of the contribution of smoking, drinking, and possibly targetable drivers of accelerated EA in older adults is essential if these tools are to be used clinically in a rational manner. Likewise, it is critical to examine these associations in a sample of Black Americans given the disproportionate burden of diabetes and other diseases of aging present in Black Americans and other marginalized groups, which may increase the likelihood of detecting associations with accelerated aging, particularly associations with cardiometabolic conditions.

Additional research is also necessary to understand the potential clinical implications of accelerated EA. One area of particular interest is smoking, which has been shown to result in EA reversion [17]. However, the reversion of smoking-related methylation changes is age- and consumption-history-dependent [23,24]. When smokers stop smoking, methylation reverts in a site-specific manner, with some sites showing more rapid reversion and others reverting slowly, if at all [17,25,26]. Because many of the loci affected by smoking are catalogued by popular existing EA indices [27,28,29], understanding the time or chronicity of smoking regarding the magnitude and breadth of changes in the total EA response would be useful. Specifically, it would be useful to know the proportion of accelerated EA that might be expected to change as a function of changes in response to abstinence.

The Promoting Strong African American Families (ProSAAF) cohort provides an ideal opportunity to generate a deeper understanding of EA indices as screening tools and their associations with medical conditions as well as cigarette and alcohol consumption [30]. Recently, we conducted health behavior interviews, performed vital sign assessments, and collected biomaterial for epigenetic and clinical laboratory analyses for 314 of the adult participants, 278 for whom we have complete data. Using these data and biomaterials, we test the relationship among objective and subjective measures of smoking and drinking, as well as self-reported diagnoses and clinically relevant biological assessments, and two commonly used EA indices, a recently developed version of the GrimAge index that incorporates the use of principal components measures to remove technical noise (PC-GrimAge), and PACE, a measure constructed using data from 19 indicators of organ system function [27,31].

## 2. Materials and Methods

This study used clinical data and biomaterials collected from the 6th wave (collected in 2021–2022) of the ProSAAF study. All procedures and protocols used in this study were approved by the University of Georgia Institutional Review Board (study title: “Protecting Strong African American Families”; IRB approval number 2012104112).

### 2.1. Study Population

The methods used to identify participants and collect earlier samples in the ProSAAF study have been described in depth elsewhere [30], and the same methods were used in the 6th wave of data collection. In brief, the sample included adult couples raising a preadolescent or early-adolescent Black child in the home at the time of the initial assessment. To be eligible, couples had to be living together, in a relationship for >2 years, and co-parenting a Black child aged 9 to 14 for >1 year. Families were recruited from low-income communities in rural Georgia between 2013 and 2014. Upon contacting the study coordinator, a full oral description and written summary of the study was provided. If all parties were willing to participate, written informed consent was obtained. In the current wave, adult participants in prior waves were contacted via phone or mail. If interested in participating, subjects were contacted by a research assistant and provided both an oral and written summary. If still interested, verbal consent was obtained and self-report data were collected. For all agreeing to biodata collection, a written informed consent form was reviewed and signed.

The clinical interview was obtained via phone or online survey tools (i.e., Qualtrics©). As part of the interview, participants provided demographic information and were asked to self-report health behaviors, health conditions, and loneliness. Binge drinking was determined by asking the question, “During the past 12 months, how often have you had a lot to drink—that is 3 or more drinks at one time?” The use of other combustible forms of tobacco-containing products was assessed by asking respondents, “How many cigars, cigarillos, filtered cigars, tobacco-filled pipes, or bowls of hookah have you smoked during the past month”? The responses for both binge drinking and smoking were binned using an ordinal scale. Self-reported health conditions were assessed by asking participants, “Has a doctor ever told you that you were suffering from any of the following chronic health problems? (0 = no, 1 = yes)”. The conditions assessed included heart disease, diabetes, cancer or leukemia, high blood pressure, arthritis or rheumatism, asthma or allergies, emphysema or chronic bronchitis, tuberculosis, circulation problems in the arms or legs, high blood sugar or PRE, ulcers of the digestive system, liver disease, kidney disease, other urinary tract disorders, anemia, stroke, allergies, thyroid or other glandular disorders, and other chronic health problems. Potential effects of exposure to community stress were assessed by asking, “How often was there a fight in which a weapon like a gun or knife was used?”, “How often was there a violent argument between neighbors?”, “How often was there a gang fight?”, “How often was there a sexual assault or rape?”, and “How often was there a robbery or mugging?” For biodata collection, a certified phlebotomist visited the home and collected 6 tubes of blood (30 mL) from each consenting participant. The phlebotomist also collected data including vital signs, height, and weight. Following collection, blood was delivered via courier overnight to the University of Iowa for further processing.

After the arrival of the specimens at the University of Iowa, DNA and sera were prepared via our standard procedures and then stored at −80 °C until used [32]. Lipid (LDL, HDL, and triglycerides) and hemoglobin A1c (HbA1c) assessments were conducted by the University of Iowa Diagnostics Laboratories under standard clinical conditions.

### 2.2. Methylation Analyses

Epigenome-wide methylation assessments were conducted using the Illumina Infinium MethylationEPIC BeadChip array (EPIC, San Diego, CA, USA) by the University of Minnesota Genome Center, according to the manufacturer’s instruction. Samples were randomized with respect to the slide and position on the arrays to minimize potential batch effects, as recommended by the Illumina Infinium Protocol Guide. The resulting data were DASEN-normalized using the MethyLumi [33], WateRmelon [34], and IlluminaHumanMethylationEPICanno.ilm10b2.hg19 [35] R packages [36]. CpG values were background-corrected using the “noob” method. Next, the data were subjected to standard sample- and probe-level quality control measures, after which data from 314 participants were retained. The principal-components-adjusted version of PC-GrimAge was estimated using the code made available on the Levine lab server (https://github.com/MorganLevineLab/PC-Clocks, accessed on 6 November 2023). Values for the PACE index were calculated using the code supplied by the developers at https://github.com/danbelsky (accessed on 8 July 2023). For comparison purposes, we also computed the most recent version of GrimAge [29] using the publicly available online tool hosted by the Horvath Lab at the University of California Los Angeles (https://dnamage.genetics.ucla.edu/ accessed on 6 November 2023), and GrimAge2 was computed using (https://dnamage.clockfoundation.org/user/login accessed on 6 November 2023). In line with common practice, for GrimAge measures, we also calculated GrimAge acceleration by subtracting the chronological age from the calculated methylation-based score. This was not used for PACE because it is already a measure of age acceleration. Finally, epigenome-wide methylation assessments were also used to provide the methylation status at a CpG locus known to be associated with diabetes in numerous clinical studies (cg19693031) [37,38,39].

The methylation-sensitive digital PCR (MSdPCR) of cg05575921 methylation (Dcg05575921) and ATS scores was conducted using the same samples of DNA and primer probe sets from Behavioral Diagnostics (Coralville, IA, USA) and both droplet digital PCR reagents and equipment from Bio-Rad (Hercules, CA, USA), according to our previously described protocols [16,32]. The status at cg05575921 is expressed as “% methylation”. ATS values are the unweighted sum of four z-scores of MSdPCR assays at four loci (cg02583484, cg04987734, cg09935388, and cg04583842) and it is a zero-centered metric in abstinent populations [20,21]. Increasing ATS values are predictive of increasing alcohol consumption, with ATS values of 3.5 and 5 being suggestive and predictive of HAC (6 or more drinks per day) [20,40].

### 2.3. Statistical Analyses

The analyses of the relationships among all behavioral health and health-related indices and the EA indices were conducted in two stages [14]. As a first step, regression was used to determine which measures were correlated with GrimAge. Next, those measures that were significantly correlated with GrimAge were each systematically tested to see if they contributed to PC-GrimAge acceleration beyond the effects of gender, age, smoking, and drinking using multiple regression modeling. As a final step, PACE was then added to the final model to see if it accounted for any additional variance.

Similarly, a parallel set of analyses was conducted using PACE, instead of GrimAge, as the outcome, again identifying individual predictors and then sets of predictors and finally adding PC-GrimAge to see if it accounted for any additional variance.

Stata 17 (College Station, TX, USA) was used at each step of the regression analyses. In each step of the model, the amount of variance explained in accelerated aging and the difference from the baseline model was determined. Akaike’s information criterion (AIC) and Bayesian information criterion (BIC) values were used to provide information about the relative fit of the competing models based on the highest R2 and minimum AIC and BIC [41,42].

## 3. Results

The demographic, physiologic, clinical, and epigenetic characteristics of the sample are provided in Table 1. Primary analyses focused on the portion of the sample with complete data (N = 278 Black Americans), the majority of whom were females (59%), with the average age being in the late 40s. Male subjects were significantly older than female subjects (*p* < 0.001). From a physiologic viewpoint, overall, the majority of participants were overweight, with the average BMI for the male and female subjects being above 30. The mean systolic blood pressure (*p* < 0.001), but not diastolic blood pressure, was substantially higher in males (140 ± 24 mm Hg) than females (127 ± 21 mm Hg, *p* < 0.001). High-density lipoprotein (LDL), but not total or low-density lipoprotein, was lower in males (50 ± 14 mg/dL) than females (55 ± 22 mg/dL, *p* < 0.02). Finally, both HbA1c (6.4 ± 1.8% mg/dL vs. 5.9 ± 1.5%, *p* < 0.02) and triglyceride levels (138 ± 79 mg/dL vs. M = 111 ± 79 mg/dL, *p* < 0.002) were higher in males than in females. Secondary sensitivity analyses utilized the full sample and employed pairwise deletion to provide results with the largest possible sample in each case. The characteristics of the pairwise samples are provided in Appendix A.

When treated as a binary variable, the rates of self-reported smoking and binge drinking (Table 1) were also significantly higher in males than in females (*p* < 0.01), which was consistent with the results for non-self-report indices of cigarette smoking (Dcg05575921) and elevated alcohol consumption (ATS). In contrast, the self-reported rates of heart disease, kidney disease, hypertension, and diabetes tended to be similar between the genders (all Chi-square values N.S.).

The GrimAge, GrimAge2, and PC-GrimAge values were also significantly higher in males than in females (all *p* < 0.0001). The overall averages for the cohorts were 47.2 ± 8.1, 55.0 ± 9.0, 61.2 ± 8.3, and 62.4 ± 7.5 years, for the chronological age, GrimAge, GrimAge2, and PC-GrimAge, respectively. The average difference between the GrimAge and PC-GrimAge was 7.4 ± 4.2 years. Despite this clear difference in average biological age, the measures were well correlated with one another (*r* values ranging from 0.879 to 0.947.)

Figure 1 shows the distribution of Alcohol T-Scores (ATS) and digital cg05575921 (Dcg05575921) values, respectively, in the subjects (n = 278). In abstinent populations, the ATS values are zero-centered and normally distributed. In this population, the ATS distribution was left-shifted but generally symmetrical (skewness, e.g., between−0.5 and 0.5). Consistent with prior findings, the ATS levels were higher in males than in females (Wilcoxon *p* < 0.0001).

Figure 2 illustrates the distribution of the Dcg05575921 values. In non-smoking adults, these values are normally distributed, with an average value of 86.6% ± 2.9% [16]. However, in response to smoking, methylation at this locus decreases in a dose-dependent manner [16]. The distribution of Dcg05575921 in these 278 subjects was significantly skewed (skewness < −0.5), reflecting the large number of smokers in the population, with the Dcg05575921 levels being substantially lower in males than females (Wilcoxon *p* < 0.0001). Overall, 40 of 164 (26%) females and 53 of 114 (46%) of males had Dcg05575921 levels of less than 80%, which is strongly predictive of daily smoking.

### 3.1. Correlations

Intercorrelations for EA indices, demographics, and all studied health and health behavior variables for the portion of the sample with complete data (N = 278) are shown in Appendix A. Age is strongly correlated with PC-GrimAge (r = 0.85) but age is not correlated with PACE (r = 0.16). In addition, PCGrimAge acceleration and PACE showed different patterns of significant correlations with many potential predictors, including, in particular, self-reported smoking and drinking and non-self-report indicators of smoking and drinking. The individual associations of predictors with PC-GrimAge acceleration can be seen in the top half of Table 2, and the associations with PACE can be seen in the top half of Table 3 (for comparison, all analyses were recalculated using pairwise deletion, yielding similar results, as can be seen in Appendix A). As can be seen in Table 2, digital assessments of smoking (Dcg055) and alcohol consumption (ATS) were significantly associated with PC-GrimAge acceleration. Also significantly associated were the methylation-based measure of diabetes (cg19693031), BMI, triglycerides, self-reported smoking, self-reported binge drinking, hypertension, arthritis, liver disease, and exposure to neighborhood crime.

### 3.2. Multiple Regression Models Predicting PC-GrimAge Acceleration

We first examined a model that included only age and sex. As can be seen in Table 2 (model 1), the baseline model of age and sex explained approximately 32.1% of the variance in PC-GrimAge acceleration. We next examined Dcg05575921 and the ATS, showing that their joint effects in an additive model (model 2) explained 48.6% of the variance in PC-GrimAge acceleration. Combining the predictors in models 1 and 2 yielded our baseline model of age, sex, dcg055, and ATS, which explained 74.4% of the variance in PC-GrimAge acceleration. We then added each of the other health-related variables that were correlated with accelerated EA when considered alone, to better gauge the extent to which PC-GrimAge acceleration would be a useful way to screen the net of the base model predictors. In particular, we sequentially added cg19693031, BMI, serum triglycerides, self-reported smoking, self-reported binge drinking, self-reported hypertension, arthritis, liver disease, and crime exposure to the model, followed by all of these together (models 4–13). None of these additions simultaneously increased the R-square and decreased both the AIC and BIC relative to the base model (model 3), suggesting that this reflected improved model performance. Finally, we added PACE both in addition to all other significant predictors and simply in addition to the base model, but these did not improve the model performance relative to the base model (model 3) either.

The results for a similar regression stepwise analysis with respect to PACE are given in Table 3. Age, Dcg05575921, and ATS (base model 3), but not sex, all significantly predict PACE. PACE was not significantly associated with g19693031 but was significantly associated with BMI, HbA1c, HDL, self-reported smoking, self-reported heart disease, hypertension, and diabetes.

### 3.3. Multiple Regression Models Predicting PACE

In contrast to the results for PCGrimAge acceleration displayed in Table 2, several predictors contributed to the variance in PACE beyond the base model of age, Dcg05575921, and ATS. In particular, BMI (model 4), HbA1c (model 5), HDL (model 6), and heart disease (model 8) were all associated with an increased R-square and decreased AIC and BIC relative to the baseline model (model 3). In addition, we considered all clinical variables contributing to the variance in PACE beyond the baseline model, more than doubling the prediction from the baseline model alone. However, the amount of variance explained by all clinical variables remained low (25.5%; see Table 3). Finally, the addition of PC-GrimAge acceleration added an additional 0.4% of variance explained to the final model with all predictors, and 1.4% to the base model (model 3), suggesting some additional processes linked to GrimAge that were relevant to PACE beyond the role of smoking and alcohol consumption.

Because 36 of the original cohort of 314 subjects had one or more missing pieces of data, the primary analyses reflect the listwise deletion of 36 subjects (final n = 278). Parallel analyses using the data from all 314 available participants are provided in the Appendix A. Please note that, in these analyses, the exact number of data points for each analysis examined may vary.

## 4. Discussion

Identifying robust, clinically actionable targets for clinicians is necessary if EA indices are to gain traction as methods to guide clinical care. In this communication, we extend our prior work to show that smoking and alcohol consumption are major drivers of GrimAge acceleration for Black American patients traversing middle age, as they are for White Americans. This suggests that PC-GrimAge may have limited value as a screening tool that can guide clinicians in choosing follow-up assessments, unless these are assessments of smoking and alcohol consumption. Conversely, PACE showed a range of significant associations with clinical predictors beyond the associations with smoking and drinking, and these were largely confined to factors related to diabetes and coronary heart disease (i.e., BMI, HbA1c, HDL, and diagnosis of heart disease). However, the amount of variance explained by these variables for PACE was low, likewise strongly limiting its value as a screening tool as well.

Our finding that 74% of all PC-GrimAge accelerated aging in these middle-aged African Americans is explained by the four-item baseline model that features objective assessments of smoking or drinking (see model 3 in Table 2). This is very similar to our prior findings in a cohort of group of 437 young African American adults from the Family and Community Health Studies [14]. Taken together with our findings in the Framingham Heart Study [43], a sample of mostly White Americans, these results suggest that the vast majority of changes in EA as assessed by GrimAge-related constructs are driven by smoking and drinking in both young and middle-aged Americans, for both Black and White Americans.

This statement should not be interpreted as stating that GrimAge indices do not predict a given condition. Indeed, in line with prior research, we found associations with cg19693031, BMI, triglycerides, self-reported smoking, self-reported binge drinking, hypertension, arthritis, liver disease, and exposure to elevated levels of community crime (see Table 2). Instead, this suggests that the variance for each of these health conditions, self-reported health behaviors, and stress exposure is likely mediated by their association with smoking or drinking. The practical implications of this for clinicians are significant. For example, it is well established that smoking is a risk factor for coronary heart disease (CHD). However, it is not the only cause of CHD, and our results suggest that GrimAge poorly captures risk factors beyond the risk conveyed by smoking or drinking. Therefore, using Carl Sagan’s old adage of “absence of evidence does not mean evidence of absence” [44], clinicians should not rely on EA indices to indicate the absence of a risk for CHD.

At the same time, these results support the contention of Faul and associates that the use of principal components reduces the “noise” in EA indices [31]. With respect to PC-GrimAge, the baseline model explained 74.4% of the variance, whereas, for the original GrimAge, the baseline model explained only 65% of the variance. However, this leads to the question as to what constitutes the remaining 25% not predicted by the model for PC-GrimAge. First, a certain portion of the remaining 25% will be noise. The use of an updated version that used a PC correction certainly improved the amount of variance predicted in our analyses. However, given the noisy nature of array assessments, considerable methylation measurement error probably remains. Second, a portion of the remaining variance will be genetic effects. When evaluated across individuals of all ancestries, up to 37% of the signal from the original GrimAge is thought to arise from heritable factors [28]. Although the genetic contribution in PC-GrimAge is not exactly known, it is likely to be considerable. Finally, a portion of the remainder of the signal is likely from common complex disorders not explained by the drinking and smoking signal. However, we note that the addition of every other clinical variable in Table 1 to the baseline model did not improve upon the baseline model. Because these other variables, such as high blood pressure, are also associated with other diseases, such as stroke or kidney failure [45], these findings suggest that the ability of GrimAge to predict disorders other than diabetes or CHD in the clinical setting may be limited as well.

The results from our regression analyses of PACE are markedly different from those for PC-GrimAge. Notably, our model building demonstrates that a portion of the variance for hypertension, heart disease, and diabetes is modestly associated with PACE, independently of the effects of smoking and drinking (see Table 3). This suggests that PACE could be a starting point for efforts to create an overall “accelerated aging” measure that could be used for broad clinical screening based on patterns of DNA methylation. PACE appears to capture signals from cardiometabolic risk factors and does so beyond the overlap with smoking and drinking. In addition, the poor correlation of the PACE values with the PC-GrimAge values (r < 0.2) and the modest improvement in variance explained achieved by adding the PC-GrimAge values to model 3 in Table 3 suggests that any portion explained by the additional variance afforded by hypertension, heart disease, and diabetes by PACE is more related to morbidity than mortality. This raising the interesting possibility of diverging measures to predict morbidity and mortality and how they might best be used clinically. This divergence makes sense because PACE was trained using data from four waves of examinations that provided “19 indicators of organ-system integrity” but not mortality data, whereas GrimAge was developed to predict mortality.

Potentially puzzling at first glance is the profound effect of adding BMI to the baseline model consisting of age, ATS, and Dcg05575921 for the prediction of PACE (model 5 in Table 3). The change in the model afforded by the addition of BMI (10.3%, see Table 3) is more than double the amount explained in the univariate analyses of BMI alone (4.3%, see Table 3). Given the strong relationship between BMI and Dcg05575921 in this cohort (r = 0.28; Appendix A) and the weak relationship of Dcg05575921 with PACE, we suspect that smoking may act as a suppressor variable, reducing the association of BMI with PACE in the overall sample. Still, smoking is the leading preventable cause of morbidity and mortality in the United States, and, after patients stop smoking, they rapidly gain weight. Because patients often quit smoking after they have developed a significant illness, such as lung cancer [46], this suggests that the incorporation of additional waves of assessments of the Dunedin subjects after they have developed smoking-related illnesses into the training of the PACE metric may lead to a differing view of the influence of smoking on organ system integrity. Furthermore, since smoking cessation is the most important preventive measure that a clinician can recommend to those who smoke, this highlights the “blind spot” of the PACE index for one of the world’s leading killers.

As in the case of our prior examinations, we note the improvement in prediction for either index when objective measures are used in place of subjective measures of substance use [14,43]. With respect to the effects of smoking, this substitution of an objective measure for self-reports results in a marked increase (from 28% to 48%) in the amount of PC-GrimAge variance explained. This increase in predictive power illustrates the potential impact of erroneous self-reports on study outcomes. For example, using the standard cutoff of 80% instead of self-reports, the number of female smokers increases from 25 to 40, while the number of male smokers increases from 34 to 53 (see Table 1). Similarly, we note that the self-report of binge drinking is not correlated with accelerated EA, while the ATS, which is tightly correlated with binge drinking, accounts for 29% of the variance. Because we and others have repeatedly noted the low reliability of self-reports of smoking, and especially drinking, in clinical studies [10,11,12,13,14,15], this suggests that investigators who are truly interested in understanding the impact of these two lifestyle factors should consider the use of biomarkers to characterize substance use in their population.

From a clinical standpoint, where does this leave a clinician who is evaluating a patient who presents at a clinic with an evaluation that shows accelerated EA? Based on these and our prior data [14], clinicians who are presented with unexpectedly high GrimAge or PC-GrimAge scores should have a strong suspicion of cigarette or alcohol consumption. Furthermore, because the rate of unreliable self-reports of these substances in clinical settings is high, we believe that clinicians should not hesitate to order biomarker assessments for tobacco and alcohol use disorders if the suspicion remains after standard clinical assessments. In contrast, for PACE, our results point to obesity, diabetes, or heart disease, as well as substance use, being potentially clinically addressable drivers, but, for these associations to guide clinical activity, it will be important to enhance their predictive validity. As Grimes and Schulz have noted, in order to be useful, screening tools must have high sensitivity and reasonable specificity [29]. However, the low correlations found by either of these indices for the common diseases of aging surveyed in our current effort suggest very limited potential for these approaches to gain the necessary positive and negative predictive value to serve as clinically useful screening tests.

In summary, in this African American cohort, beyond addressing smoking, drinking, and weight, our findings do not offer strong support for EA indices in directing any other assessments besides those routinely obtained. In light of prior findings by others as to the consistency of EA index results [2], we suggest that clinicians presented with outside EA index results by patients view them as evidence of engagement with an understanding that much of their signal may result from substance use (in the case of GrimAge) or cardiometabolic risk and substance use (in the case of PACE) and then proceed with a balanced assessment strategy based on their objective and subjective observations.

## Figures and Tables

**Figure 1 genes-15-00869-f001:**
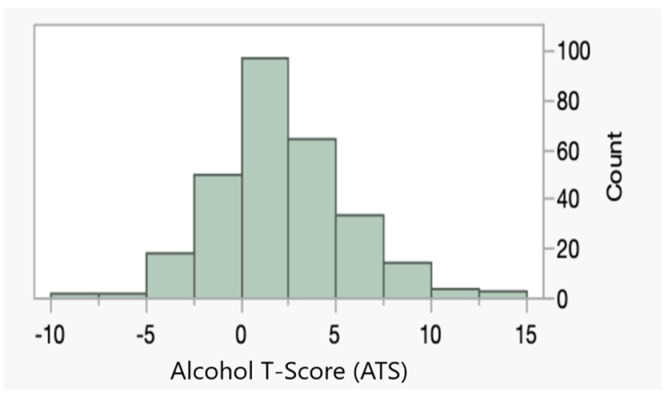
The distribution of Alcohol T-Scores (ATS) in the 278 subjects with complete data. The ATS is a unitless zero-ordered metric in abstinent populations.

**Figure 2 genes-15-00869-f002:**
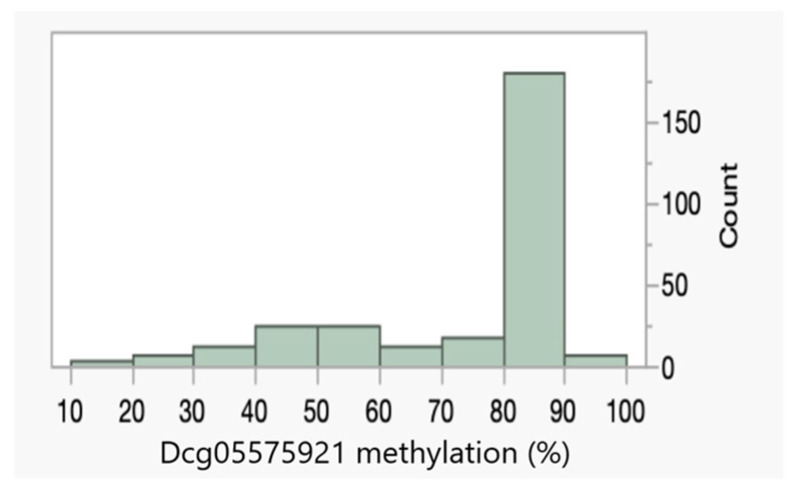
The distribution of digital cg05575921 (Dcg055759231) methylation assessments in the 278 subjects with complete data. In non-smoking adult subjects, methylation is 86.6% ± 2.9%. In response to smoking, methylation decreases in a dose-dependent manner.

**Table 1 genes-15-00869-t001:** Clinical and demographic characteristics of participants with all data (N = 278).

	Female	Male
Sex	164	114
Age (chronological)	46.1 ± 7.1	49.6 ± 9.3
**Physiologic Parameters**		
BMI **	35.4 ± 8.1	32 ± 8.2
Systolic BP **	127 ± 21 mm Hg	140 ± 24 mm Hg
Diastolic BP **	87 ± 12 mm Hg	88 ± 14 mm Hg
Cholesterol	182 ± 40 mg/dL	179 ± 41 mg/dL
LDL **	106 ± 35 mg/dL	102 ± 38 mg/dL
HDL **	55 ± 22 mg/dL	49 ± 14 mg/dL
HbA1c **	5.9 ± 1.5%	6.4 ± 1.8%
Triglycerides **	111 ± 60 mg/dL	138 ± 79 mg/dL
**Self-Reported Behaviors and Conditions**		
Smoking *	25 (15%)	34 (30%)
Binge Drinking *	33 (20%)	42 (37%)
Heart Disease	15 (9%)	17 (15%)
Hypertension	89 (54%)	64 (56%)
Diabetes	29 (18%)	25 (22%)
Arthritis	50 (31%)	24 (21%)
Cancer	2 (1%)	3 (3%)
Liver Disease	3 (2%)	1 (1%)
Kidney Disease	6 (4%)	5 (4%)
Cataracts	7 (4%)	8 (7%)
**Exposure to Community Crime**		
Crime	0.103 ± 0.25	0.174 ± 0.30
**Epigenetic Measures of Age and Aging**		
GrimAge	52.8 ± 8.8 years	58.9 ± 8.4 years
GrimAge2	59.0 ± 7.2 years	64.4 ± 8.7 years
PCGrimAge	60.1 ± 6.2 years	66.2 ± 7.9 years
GrimAgeAcc *	6.7 ± 7.7 years	9.35 ± 5.7 years
GrimAge2Acc *	13.0 ± 5.7 years	14.9 ± 6.5 years
PCGrimAge Acc *	14.1 ± 3.9 years	16.6 ± 4.3 years
PACE	1.07 ± 0.17	1.08 ± 0.14
Dcg05575921	79 ± 15%	68 ± 21%
ATS	1.5 ± 2.9	3.2 ± 3.7
Cg19693031	78.3%	73.1%

Note: Numbers to the right of the “±” represent the standard deviation. BMI = body mass index; systolic BP = systolic blood pressure; diastolic BP = diastolic blood pressure; LDL = low-density lipoprotein; HDL = high-density lipoprotein; HbA1c = glycated hemoglobin; triglycerides = triglycerides are a type of fat found in the blood. GrimAgeAcc = acceleration of Grim age is determined by subtracting chronological age from methylation-based age. GrimAge2Acc = Grim Age 2 Based on Real Age is determined by subtracting chronological age from methylation-based age. PCGrimAge Acc = principal component Grim Age acceleration is determined by subtracting chronological age from methylation-based age. * = *p* < 0.05, ** *p* < 0.001.

**Table 2 genes-15-00869-t002:** Regression modeling predicting PCGrimAge accelerated aging using listwise deletion of those with missing predictors (N = 278).

		Adj. R2	AIC	BIC
**Demographic**	Age	**0.168**	1548	1555
	Sex	**0.085**	1574	1581
**Epigenetic**	Dcg05575921 (Dcg055)	**0.459**	1428	1435
	ATS	**0.285**	1505	1513
	cg19693031	**0.037**	1588	1596
**Vitals**	BMI	**0.015**	1594	1602
	Systolic	0.004	1598	1605
	Diastolic	−0.003	1600	1607
**Serum**	HbA1c	−0.001	1599	1606
	Cholesterol	0.004	1598	1604
	LDL	0.006	1597	1604
	HDL	0.005	1597	1604
	Triglycerides	**0.015**	1595	1601
**Med History**	Smoking	**0.257**	1516	1524
	Binge Drinking	**0.055**	1583	1591
	Heart Disease	−0.002	1599	1606
	Hypertension	**0.043**	1587	1593
	Diabetes	−0.003	1600	1607
	Arthritis	**0.024**	1592	1599
	Cancer	−0.002	1599	1606
	Liver Disease	**0.013**	1595	1602
	Kidney Disease	0.004	1598	1605
	Cataracts	0.005	1597	1605
**Crime**	Crime	**0.025**	1592	1599
**Model**				
1	Age + Sex	**0.321**	1492	1503
2	Dcg055 + ATS	**0.486**	1415	1426
3	Age + Sex + Dcg055 + ATS	**0.744**	1223	1241
4	Model 3 + cg19693031	**0.747**	1221	1242
5	Model 3 + BMI	0.744	1224	1246
6	Model 3 + Triglycerides	0.745	1223	1245
7	Model 3 + Smoking	0.743	1225	1247
8	Model 3 + Binge Drinking	0.743	1225	1247
9	Model 3 + Hypertension	0.743	1225	1247
10	Model 3 + Arthritis	0.744	1225	1246
11	Model 3 + Liver Disease	0.743	1225	1247
12	Model 3 + Crime	**0.748**	1220	1242
13	Model 3 + All Significant Predictors	**0.748**	1216	1241
14	Model 3 + All Significant Predictors + PACE	0.752	1217	1246
15	Model 3 + PACE	0.744	1224	1246

Note: Univariate regression results whose significance is *p* < 0.05 are indicated by bolded Adj. R2. Multivariate regression results are bolded when the added variable is significant at *p* < 0.05 in the context of the other variables. AIC = Akaike’s information criterion; BIC = Bayesian information criterion (lower scores indicate better fit).

**Table 3 genes-15-00869-t003:** Regression modeling predicting PACE using listwise deletion of those with missing predictors (N = 278).

		Adj. R2	AIC	BIC
**Demographic**	Age	**0.022**	−248	−241
	Sex	−0.002	−241	−234
**Epigenetic**	Dcg05575921 (Dcg055)	**0.036**	−252	−245
	ATS	**0.103**	−272	−265
	cg19693031	0.003	−243	−235
**Vitals**	BMI	**0.043**	−254	−247
	Systolic	0.006	−243	−236
	Diastolic	0.003	−242	−235
**Serum**	HbA1c	**0.039**	−253	−246
	Cholesterol	0.008	−244	−237
	LDL	−0.004	−241	−234
	HDL	**0.079**	−265	−257
	Triglycerides	0.007	−244	−236
**Med History**	Smoking	**0.014**	−246	−239
	Binge Drinking	−0.000	−242	−234
	Heart Disease	**0.047**	−255	−248
	Hypertension	**0.012**	−245	−238
	Diabetes	**0.022**	−248	−241
	Arthritis	0.004	−243	−236
	Cancer	−0.001	−241	−234
	Liver Disease	−0.003	−241	−233
	Kidney Disease	0.006	−243	−236
	Cataracts	−0.003	−241	−233
**Crime**	Crime	0.003	−243	−235
**Model**				
1	Age	**0.022**	−248	−241
2	Dcg055 + ATS	**0.100**	−270	−259
3	Age + Dcg055 + ATS	**0.104**	−270	−256
4	Model 3 + BMI	**0.207**	−303	−285
5	Model 3 + HbA1c	**0.129**	−277	−259
6	Model 3 + HDL	**0.179**	−294	−276
7	Model 3 + Smoking	0.101	−268	−250
8	Model 3 + Heart Disease	**0.126**	−276	−258
9	Model 3 + Hypertension	0.111	−271	−253
10	Model 3 + Diabetes	**0.116**	−273	−255
11	Model 3 + All Significant Predictors *	**0.255**	−318	−285
12	Model 3 + All Significant + PCGrimAgeAcc	0.258	−317	−280
13	Model 3 + PCGrimAgeAcc	0.118	−269	−250

Note: Univariate regression results whose significance is *p* < 0.05 are indicated by bolded Adj. R2. Multivariate regression results are bolded when the added variable is significant at *p* < 0.05 in the context of the other variables. AIC = Akaike’s information criterion; BIC = Bayesian information criterion (lower scores indicate better fit). * = BMI, HbA1c, HDL, Heart Disease, and Diabetes.

## Data Availability

The genome-wide data included in this manuscript were prepared with funding from the United States National Institutes of Health (NIH). They are not yet publicly available but are being prepared for submission consistent with NIH Genomic Data Sharing policies.

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
