# Peer review of "Objective Assessments of Smoking and Drinking Outperform Clinical Phenotypes in Predicting Variance in Epigenetic Aging"

_genes, 2024, doi:10.3390/genes15070869_

Round 1

Reviewer 1 Report

Comments and Suggestions for Authors

In the manuscript entitled "Digital Assessments of Smoking and Drinking are more explanatory than Clinical Phenotypes for Variance in PC-GrimAge but not for DunedinPACE", Philibert and co-authors presented an analysis of the relationship between digital assessments of smoking and drinking behaviors, and epigenetic aging measures like PC-GrimAge and DunedinPACE. The central concept of the manuscript is engaging. Nonetheless, there are several significant and minor concerns that require resolution.

Major issues:

·   The introduction section is overly extensive. Kindly condense it.

·   The Materials and Methods section should primarily focus on the models; however, only the final two paragraphs address this, and they are difficult to comprehend without first reviewing the results. Please revise these paragraphs to enhance clarity.

·   Certain descriptions in the results section do not correspond with the data shown in the tables. Please review and amend these discrepancies.

·   Please direct key sentences in your discussion section back to your results section.

Minor issues:

·   Line 18: Please include the abbreviation (PACE) immediately following the first mention of DunedinPACE, and then refer to it as PACE throughout the remainder of the paper.

·       Lines 82 to 85: Kindly consider revising this sentence for clarity.

·       Line 145: Why have you used "both" in this context? Are you referring to providing summaries in both oral and written forms?

·       Line 164: Please put “.” after “problems”.

·       Lines 209 to 219: This paragraph is difficult to comprehend. To fully grasp its content, one should refer to the results section. Kindly consider revising this paragraph for clarity.

·       Table 1: Please specify the meaning you intend to convey with each row. What is on the right side of ± (SD or SE)?  Please also put an extra column for p-values to show if there is a significant difference between male and female for each row. What are * and ** in the first column? It is good to write (chronological) in front of Age as “Age (chronological).

·       Lines 254 to 257: unlike what you wrote, the value for cigarette smoking (Dcg05575921) is lower in males.

·       Line 255: You mentioned P<0.01 here and you should show that in table 1 as well.

·       Line 265: with what?

·   In figure 2 legend, you wrote “In non-smoking adult subjects, methylation is 86.6% ± 2.9%. In response to smoking, methylation decreases in a dose dependent manner.” but I do not think these 2 sentences are depicted in Figure 2. Correct?

·   Line 275: 178 females + 124 males = 302. I do not understand how these numbers are coming from. In MM, you mentioned “Please note that because complete serological, self-report or methylation data were not available for all 314 subjects, the primary analyses reflect list-wise deletion of 36 subjects (final n=278).”. So please explain what is 302?

·   Line 285: You wrote “PC-GrimAge (r = .85)” while r is  

·   Table 2, models 12 and 13: What is the meaning of *?  What is the meaning of all significant predictors?

·   Table 2, model 13: move down number 13.

·   Lines 319 and 322: I think you meant “baseline model (Model 3)”, not “base model (Model 3)” as you defined Model 1 as your base model in line 307.

·   Lines 323 to 327: Put it as the first paragraph of section “3.3. Multiple regression models predicting DunedinPACE”.

·   Line 324: you mentioned “but not sex” but adj. R2 for sex is bolded.

·   Line 325: Table 3 shows that PACE is significantly associated with g19693031. Look at the bolded value in Table 3.

·   Line 326: Values for A1c, heart disease, and diabetes are not bolded in Table 3.

·   Line 336: I think your baseline model (Model 3) also has sex. Correct?

·   Line 3337: BMI is model 5. I do not see HbA1c (model 5), HDL (model 6), and heart disease (model 8) in Table 3!!!

·   Table 2 and 3: Could you verify whether the R² values for all models numbered 5-11 and 14-15 should remain unbolded?

·   Line 339 to 346: Kindly consider revising this sentence (beginning with “In addition”) for clarity. How did you get 25.5%? You mentioned “Finally, the addition of PC-GrimAge acceleration added an additional 0.4% of variance explained to the final model with all predictors” but PC-GrimAge does not appear to be included in Model 3 as shown in Table 3. Please update Table 3 accordingly.

·       Line 334: baseline model (model 3), not “base model (Model 3)”.

·       Line 352: Please put a citation after “White Americans”.

·       Line 376 to 377: Please direct this sentence, which starts with "However," back to your results section.

·       Lines 382 and 383 and 395: baseline model (model 3), not “base model (Model 3)”.

·       Line 385: Please put “.” after “PC-GrimAge”.

·   Line 409 to 410: PC-GrimAge does not appear to be included in Model 3 as shown in Table 3. Please update Table 3 accordingly.

·   Line 418: Are you missing Sex from your baseline model here?

·   Lines 419 and 420: Did you mention 10.3% and 4.3% in the results section?

·       Lines 438 to 440: Please direct this sentence, which starts with "For example," back to your results section.

Author Response

Review 1.

Major issues:

Comment: The introduction section is overly extensive. Kindly condense it. 

Response: We reduced the discussion by 22 lines of text.

Comment: The Materials and Methods section should primarily focus on the models; however, only the final two paragraphs address this, and they are difficult to comprehend without first reviewing the results. Please revise these paragraphs to enhance clarity.

Response:  We have rewritten the section as per the reviewer’s request.

Comment: Certain descriptions in the results section do not correspond with the data shown in the tables. Please review and amend these discrepancies.

Response:  That is because Table 2 was pasted overtop the original Table 3.  We have fixed that problem. Now, the results look correct.  Thank you for catching that embarrassing error.

Comment: Please direct key sentences in your discussion section back to your results section.

Response:  Absolutely.  We have added text to refer them back to the appropriate figure or table.

Minor Issues:

All were addressed.  A sizeable number of these arose simply because I pasted Table 2 overtop the original Table 3.

Reviewer 2 Report

Comments and Suggestions for Authors

The article is well written, the methodology and statistics are adequate, the objectives are interesting, as is the assessment of epigenetic parameters, and it focuses on an understudied population. So I have only a few suggestions that I would like to see considered.

Overall, I found the article difficult to read and to some extent tedious due to the repetition of statistics and table values in the text. I suggest highlighting the main results at the beginning of each section and reducing the number of data that appear in the text and are already in the tables.   

The title of the article is difficult to understand and not very interesting for researchers not specialized in the subject, which limits its scope, I suggest restructuring it.

In materials and methods, I suggest dividing it into sections: study population, interview, methylation analysis, statistical analysis, etc. 

Indicate whether the tables report mean +- standard deviation or some other statistic. 

Figure 1 and Figure 2 could be shown as one figure.

Justify the inclusion of Figures 1 and 2, how the distribution of indices relates to the objectives.

Comments on the Quality of English Language

Moderate editing of English language required

Author Response

Reviewer 2.

Major issues:

Comment: Overall, I found the article difficult to read and to some extent tedious due to the repetition of statistics and table values in the text. I suggest highlighting the main results at the beginning of each section and reducing the number of data that appear in the text and are already in the tables.  

Response: This is difficult to do without making a combined results/discussion section.  Still, we have made modest changes to the results structure that should help address the reviewer’s concern.

Comment: The title of the article is difficult to understand and not very interesting for researchers not specialized in the subject, which limits its scope, I suggest restructuring it.

Response: Very much agreed.  We have re-titled the article to grab the reader’s attention.

Comment: In materials and methods, I suggest dividing it into sections: study population, interview, methylation analysis, statistical analysis, etc.

Response: Done per the reviewer’s request.

Comment: Indicate whether the tables report mean +- standard deviation or some other statistic.

Response: Done per the reviewer’s request.